# *fruitless* tunes functional flexibility of courtship circuitry during development

Jie Chen[1], Sihui Jin[1], Dandan Chen[1], Jie Cao[1], Xiaoxiao Ji[1], Qionglin Peng[1]*, Yufeng Pan[1,2]*

[1]The Key Laboratory of Developmental Genes and Human Disease, School of Life Science and Technology, Southeast University, Nanjing, China; [2]Co-innovation Center of Neuroregeneration, Nantong University, Nantong, China

**Abstract** Drosophila male courtship is controlled by the male-specific products of the *fruitless* (*fru^M*) gene and its expressing neuronal circuitry. *fru^M* is considered a master gene that controls all aspects of male courtship. By temporally and spatially manipulating *fru^M* expression, we found that *fru^M* is required during a critical developmental period for innate courtship toward females, while its function during adulthood is involved in inhibiting male–male courtship. By altering or eliminating *fru^M* expression, we generated males that are innately heterosexual, homosexual, bisexual, or without innate courtship but could acquire such behavior in an experience-dependent manner. These findings show that *fru^M* is not absolutely necessary for courtship but is critical during development to build a sex circuitry with reduced flexibility and enhanced efficiency, and provide a new view about how *fru^M* tunes functional flexibility of a sex circuitry instead of switching on its function as conventionally viewed.

*For correspondence:
pengqionglin@seu.edu.cn (QP);
pany@seu.edu.cn (YP)

Competing interests: The authors declare that no competing interests exist.

## Introduction

Drosophila male courtship is one of the best understood innate behaviors in terms of genetic and neuronal mechanisms (*Dickson, 2008*; *Yamamoto and Koganezawa, 2013*). It has been well established that the *fruitless* (*fru*) gene and its expressing neurons control most aspects of such innate behavior (*Ito et al., 1996*; *Manoli et al., 2005*; *Ryner et al., 1996*; *Stockinger et al., 2005*). The male-specific products of the P1 promoter of the *fru* gene (*fru^M*) are expressed in ~2000 neurons, which are inter-connected to form a sex circuitry from sensory neurons to motor neurons (*Cachero et al., 2010*; *Lee et al., 2000*; *Manoli et al., 2005*; *Stockinger et al., 2005*; *Usui-Aoki et al., 2000*; *Yu et al., 2010*). *fru^M* function is necessary for the innate courtship behavior and sufficient for at least some aspects of courtship (*Baker et al., 2001*; *Demir and Dickson, 2005*; *Manoli et al., 2005*). Thus, the study of *fru^M* function in controlling male courtship serves as an ideal model to understand how innate complex behaviors are built into the nervous system by regulatory genes (*Baker et al., 2001*).

Although *fru^M* serves as a master gene controlling Drosophila male courtship, we recently found that males without *fru^M* function, although did not court if raised in isolation, were able to acquire at least some courtship behaviors if raised in groups (*Pan and Baker, 2014*). Such *fru^M*-independent but experience-dependent courtship acquisition requires another gene in the sex determination pathway, the *doublesex* (*dsx*) gene (*Pan and Baker, 2014*). *dsx* encodes male- and female-specific DSX proteins (DSX^M and DSX^F, respectively) (*Burtis and Baker, 1989*), and DSX^M is expressed in ~700 neurons in the central nervous system (CNS), the majority of which also express *fru^M* (*Rideout et al., 2010*; *Robinett et al., 2010*). It has been found that the *fru^M* and *dsx^M* co-expressing neurons are required for courtship in the absence of *fru^M* function (*Pan and Baker, 2014*). Thus *fru^M*-expressing neurons, especially those co-expressing *dsx^M*, control the expression of courtship behaviors even in the absence of FRU^M function. Indeed, although the gross neuroanatomical

**eLife digest** Innate behaviors are behaviors that do not need to be learned. They include activities such as nest building in birds and web spinning in spiders. Another behavior that has been extensively studied, and which is generally considered to be innate, is courtship in fruit flies. Male fruit flies serenade potential mates by vibrating their wings to create a complex melody. This behavior is under the control of a gene called '*fruitless*', which gives rise to several distinct proteins, including one that is unique to males. For many years, this protein – called Fru$^M$ – was thought to be the master switch that activates courtship behavior.

But recent findings have challenged this idea. They show that although male flies that lack Fru$^M$ fail to show courtship behaviors if raised in isolation, they can still learn them if raised in groups. This suggests that the role of Fru$^M$ is more complex than previously thought. To determine how Fru$^M$ controls courtship behavior, Chen et al. have used genetic tools to manipulate Fru$^M$ activity in male flies at different stages of the life cycle and distinct cells of the nervous system.

The results revealed that Fru$^M$ must be present during a critical period of development – but not adulthood – for male flies to court females. However, Fru$^M$ strongly influences the type of courtship behavior the male flies display. The amount and location of Fru$^M$ determines whether males show heterosexual, homosexual or bisexual courtship behaviors. Adult flies with lower levels of Fru$^M$ show an increase in homosexual courtship and a decrease in heterosexual courtship.

These findings provide a fresh view on how a master gene can generate complex and flexible behaviors. They show that *fruitless*, and the Fru$^M$ protein it encodes, work distinctly at different life cycles to modify the type of courtship behavior shown by male flies, rather than simply switching courtship behavior on and off. Exactly how Fru$^M$ acts within the fruit fly brain to achieve these complex effects requires further investigation.

features of the *fru$^M$*-expressing circuitry are largely unaffected by the loss of *fru$^M$* (*Manoli et al., 2005*; *Stockinger et al., 2005*), detailed analysis revealed morphological changes of many *fru$^M$*-expressing neurons (*Cachero et al., 2010*; *Kimura et al., 2005*; *Kimura et al., 2008*; *Mellert et al., 2010*). Recent studies further reveal that FRU$^M$ specifies neuronal development by recruiting chromatin factors and changing chromatin states, and also by turning on and off the activity of the transcription repressor complex (*Ito et al., 2012*; *Ito et al., 2016*; *Sato et al., 2019a*; *Sato et al., 2019b*; *Sato and Yamamoto, 2020*).

That FRU$^M$ functions as a transcription factor to specify development and/or physiological roles of certain *fru$^M$*-expressing neurons, and perhaps the interconnection of different *fru$^M$*-expressing neurons to form a sex circuitry raises important questions regarding when *fru$^M$* functions and how it contributes to the sex circuitry (e.g., how the sex circuitry functions differently with different levels of FRU$^M$), especially in the background that *fru$^M$* is not absolutely necessary for male courtship (*Pan and Baker, 2014*). To at least partially answer these questions, we temporally or spatially knocked down *fru$^M$* expression and compared courtship behavior in these males with that in wild-type males or *fru$^M$* null males and revealed crucial roles of *fru$^M$* during a narrow developmental window for the innate courtship toward females. We also found that the sex circuitry with different *fru$^M$* expression has distinct function such that males could be innately heterosexual, homosexual, bisexual, or without innate courtship but could acquire such behavior in an experience-dependent manner. Thus, *fru$^M$* tunes functional flexibility of the sex circuitry instead of switching on its function as conventionally viewed.

## Results

### *fru$^M$* is required during pupation for regular neuronal development and female-directed courtship

To specifically knockdown *fru$^M$* expression, we used a microRNA targeting *fru$^M$* (*UAS-fruMi* at attp2 or attp40) and a scrambled version as a control (*UAS-fruMiScr* at attp2) as previously used (*Chen et al., 2017*; *Meissner et al., 2016*). Driving the *fru$^M$* microRNA by *fru$^{GAL4}$* specifically knocked down mRNA of *fru$^M$*, but not the common form of *fru* (*Figure 1—figure supplement 1A–*

*C*). We firstly tested male courtship without food in the behavioral chamber. Knocking down *fru^M* in all the *fru^GAL4*-labeled neurons eliminated male courtship toward females (courtship index [CI], which is the percentage of observational time that males displayed courtship, is nearly 0) (*Figure 1A*), consistent with previous findings that *fru^M* is required for innate male–female courtship (*Demir and Dickson, 2005*; *Pan and Baker, 2014*). As *fru^GAL4* drives expression throughout development and adulthood (*Figure 1—figure supplement 1D–K*), we set out to use a temperature-dependent *tub-GAL80^ts* transgene to restrict *UAS-fruMi* expression (e.g., at 30°C) at different developmental stages. We raised *tub-GAL80 ^ts/+*; *fru^GAL4/UAS-fruMi* flies at 18°C (permissive for GAL80^ts that inhibits GAL4 activity) and transferred these flies to fresh food vials every 2 days. In this way, we generated *tub-GAL80 ^ts/+*; *fru^GAL4/UAS-fruMi* flies at nine different stages from embryos to adults and incubated all flies at 30°C to allow *fru^M* knockdown for 2 days, then placed all flies back to 18°C until courtship test (*Figure 1B*). We found that males with *fru^M* knocked down at stage 5 for 2 days, matching the pupation phase, rarely courted (CI < 10%), and none successfully mated, while males with *fru^M* knocked down near this period (stages 4 and 6) showed a partial courtship or mating deficit, and males with *fru^M* knocked down at earlier or later stages showed strong courtship toward females and successful mating (*Figure 1C,D*).

To validate efficiency of *fru^M* knockdown during specific developmental periods, we generated an antibody against Fru^M as well as a V5 knock-in into the *fru* gene (*fru^V5*) to visualize Fru^M expression. Both tools successfully labeled male-specific Fru^M proteins (*Figure 1—figure supplement 2*), and there is almost perfect overlap of the two markers (*Figure 1E,G*). Note that the anti-Fru^M antibody also labeled several pairs of false-positive neurons in both wild-type and *fru^M* mutants (*Figure 1—figure supplement 2*), indicating the strong but not perfect specificity of this antibody (*Figure 1—figure supplement 2B–D*). To test whether 2 day heat shock at 30°C is sufficient to knockdown *fru^M* expression, we dissected brains of *tub-GAL80^ts/UAS-fruMi*; *fru^GAL4/fru^V5* males immediately after 2 day heat shock at stage 5 or 7 and found that anti-V5 and anti-Fru^M signals were both dramatically decreased, such that only a small fraction of neurons could be weakly labeled; in contrast, control males with the same age but raised at 18°C have regular anti-V5 and anti-Fru^M signals (*Figure 1E–H*). These results indicate that induction of *fru^M* microRNA during development for 2 days could effectively knockdown *fru^M* expression.

As induced *fru^M* microRNA may not be degraded immediately and has longer effect, we further tested to how much extent such knockdown effect may last. Thus, we dissected brains of adult *tub-GAL80^ts/UAS-fruMi*; *fru^GAL4/fru^V5* males that have been heat shocked for 2 days at different developmental stages (from stages 1 to 9) and found that males that have been heat shocked at earlier stages (from stages 1 to 5) still have strong Fru^M expression (*Figure 1—figure supplement 3A–F*), suggesting effective restore of Fru^M expression after transferring at 18°C. However, males that have been heat shocked at later stages (stages 6–9) have obviously reduced Fru^M expression (*Figure 1—figure supplement 3G–J*), suggesting a partial restore of Fru^M expression, probably due to prolonged *fru^M* microRNA effect. Note that knocking down *fru^M* expression at these later stages has partial (stage 6) or no effect (other stages) on male courtship, comparing with *fru^M* knockdown at stage 5 that almost eliminated male courtship. Together these results indicate a critical developmental period during pupation (from late larvae at stage 5 to early pupas at stage 6) where *fru^M* is required for adult male courtship toward females.

We reasoned that *fru^M* function during pupation may be involved in neuronal development for circuit construction. Thus we set out to examine the morphology of a subset of *fru^M*-positive gustatory receptor neurons (GRNs) innervating the ventral nerve cord (VNC) in *tub-GAL80^ts/UAS-mCD8GFP*; *fru^GAL4/UAS-fruMi* males that have been heat shocked for 2 days in different developmental stages, as it has been found that *fru^M* is required for the male-specific midline crossing of these GRNs (*Mellert et al., 2010*). We found that these GRNs were only labeled in males that have been heat shocked after stage 4, probably because these GRNs were developed after stage 4 (*Figure 1—figure supplement 4A–C*), consistent with a previous study (*Mellert et al., 2012*). Interestingly, we found that all males heat shocked at stage 5 for 2 days showed defect of midline crossing in these GRNs, and 60% of males heat shocked at stage 6 for 2 days showed defect of midline crossing, while all males heat shocked after stage 6 showed regular midline crossing (*Figure 1—figure supplement 4C,D*). Males heat shocked for 4 days during adulthood also have regular midline crossing (*Figure 1—figure supplement 4C,D*). These results clearly showed a critical developmental period

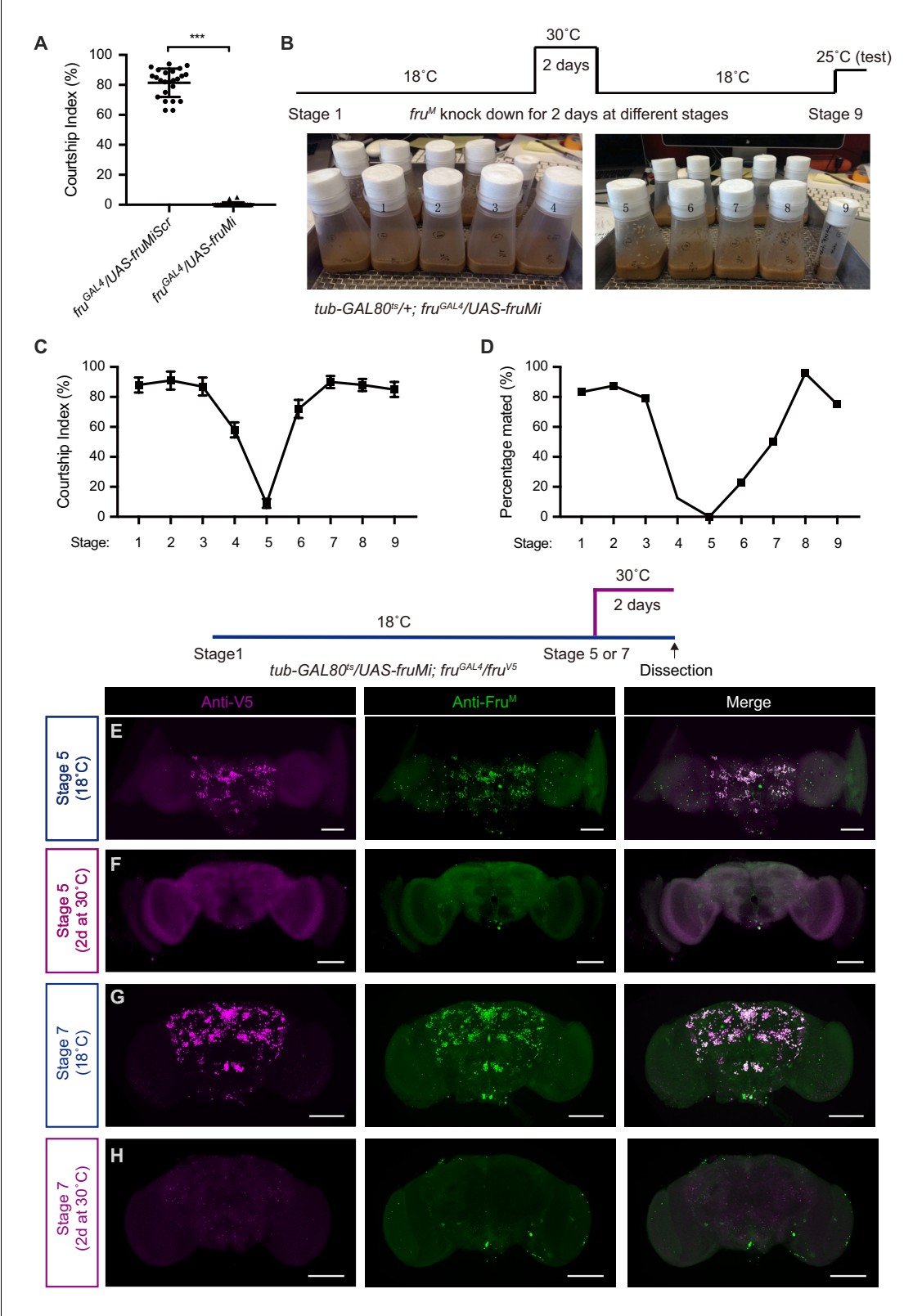

**Figure 1.** *fru^M* is required during pupation for female-directed courtship in adult males. (**A**) Knocking down *fru^M* using RNAi throughout development and adulthood eliminated male courtship toward virgin females. n = 24 for each. ***p<0.001, unpaired t-test. (**B**) A schematic of genetic strategy to knockdown *fru^M* at different developmental stages for 2 days. Stages 1–9 refer to specific developmental stages from embryos to newly eclosed adults with interval of 2 days. (**C** and **D**) Courtship indices of males with *fru^M* knocked down at specific developmental stages as indicated above toward virgin

*Figure 1 continued on next page*

*Figure 1 continued*

females. Males with *fru*$^M$ knocked down at stage 5 for 2 days (a period of pupation from stage 5 to 6, see above picture) rarely courted virgin females (C), and none successfully mated (D). Knocking down *fru*$^M$ at stages near 5 (e.g., stage 4 or 6) also partially impairs courtship and mating success. Knocking down *fru*$^M$ at earlier or later stages has no obvious effect on courtship and mating. n = 24 for each. Error bars indicate SEM. (E–H) Two day heat shock at 30°C effectively knocks down *fru*$^M$ expression during development. Anti-V5 and anti-Fru$^M$ signals are dramatically decreased after heat shock at stage 5 (E and F) or 7 (G and H) in *tub-GAL80*$^{ts}$/*UAS-fruMi*; *fru*$^{GAL4}$/*fru*$^{V5}$ males. Scale bars, 100 μm. Representative of five samples each.

The online version of this article includes the following source data and figure supplement(s) for figure 1:

**Source data 1.** Source data for *Figure 1*.
**Figure supplement 1.** *fru*$^M$ microRNA efficiency and *fru*$^M$ expression patterns across development.
**Figure supplement 2.** Validation of anti-Fru$^M$ antibody and the *fru*$^{V5}$.
**Figure supplement 3.** Adult *fru*$^M$ expression after 2 day induction of *fru*$^M$ microRNA during development.
**Figure supplement 4.** *fru*$^M$ is required during a specific developmental period for regular neuronal development.

during pupation where Fru$^M$ functions to ensure regular development of GRNs and enable innate male courtship toward females.

## *fru*$^M$ function during adulthood inhibits male–male courtship

As knocking down *fru*$^M$ at stage 9 when flies were newly eclosed did not affect male courtship (CI > 80%) and mating success (*Figure 1C,D*), we further tested the role of *fru*$^M$ in adulthood using different approaches. We set out to express the female-specific *transformer* (*traF*) gene (*Baker and Ridge, 1980*; *McKeown et al., 1988*) to feminize all *fru*$^{GAL4}$ labeled neurons, in addition to the above *fru*$^M$ RNAi experiments. We express *UAS-traF* or *UAS-fruMi* in all the *fru*$^{GAL4}$-labeled neurons specifically during adulthood for 4 days before test (see procedure above each figure) for single-pair male–female, male–male, and male chaining (in groups of eight males) behaviors. We found that overexpression of *traF* in all *fru*$^{GAL4}$ labeled neurons during adulthood for 4 days did not affect male–female courtship (*Figure 2A*), but slightly increased male–male (*Figure 2B*) and male chaining behaviors (*Figure 2C*). Furthermore, knocking down *fru*$^M$ in all *fru*$^{GAL4}$-labeled neurons during adulthood for 4 days did not affect male–female (*Figure 2A*) or male–male courtship (*Figure 2B*), but significantly increased male chaining behaviors (*Figure 2C*). We also checked Fru$^M$ expression in males that have been heat shocked for 4 days during adulthood using anti-V5 and anti-Fru$^M$ antibodies, and found that Fru$^M$ expression was almost eliminated, while control males have regular Fru$^M$ expression (*Figure 2D,E*). These results indicate that although *fru*$^M$ function during adulthood is dispensable for female-directed courtship, it is involved in inhibiting male–male courtship behaviors. Thus, Fru$^M$ has distinct functions during development and adulthood for male courtship behaviors.

## *fru*$^M$ expression determines courtship modes

The above results indicate crucial roles of *fru*$^M$ during pupation for female-directed courtship in adult males. We reasoned that *fru*$^M$ function during pupation may specify the construction of courtship circuitry and affects female-directed courtship as well as other courtship behaviors, especially given our previous findings that *fru*$^M$ null males were able to acquire courtship behavior after group-housing (*Pan and Baker, 2014*). Thus, we set out to compare courtship behaviors in males with distinct *fru*$^M$ expression modes, such as with wild-type *fru*$^M$, systemic low level of *fru*$^M$, spatially low level of *fru*$^M$, or completely without *fru*$^M$ function. We tested one-time single-pair male–female and male–male courtship (single housed before test) as well as male chaining in groups of eight males over 3 days on food for better comparison of these courtship assays, as courtship by *fru*$^M$ null males largely depends on food presence (*Pan and Baker, 2014*). We found that male–male courtship in *fru*$^M$ knocked down males is higher if tested on food, consistent with a courtship promoting role by food (*Grosjean et al., 2011*; *Pan and Baker, 2014*), while courtship in wild-type males on food or without food is not changed in our assays (*Figure 3—figure supplement 1*). We found that wild-type males performed intensive courtship behavior toward virgin females (CI > 80%) and rarely courted males (CI ~0) (*Figure 3A*). Furthermore, these control males did not show any chaining behavior after grouping from 3 hr to 3 days (ChI = 0) (*Figure 3B*). In striking contrast, *fru*$^M$ null mutant males rarely courted either females or males (*Figure 3C*, *Figure 3—figure supplement 2A, C, and E*); however, these males developed intensive chaining behavior after grouping for 1–3 days (*Figure 3D*, *Figure 3—figure supplement 2B, D, and F*). These observations replicated previous findings that there

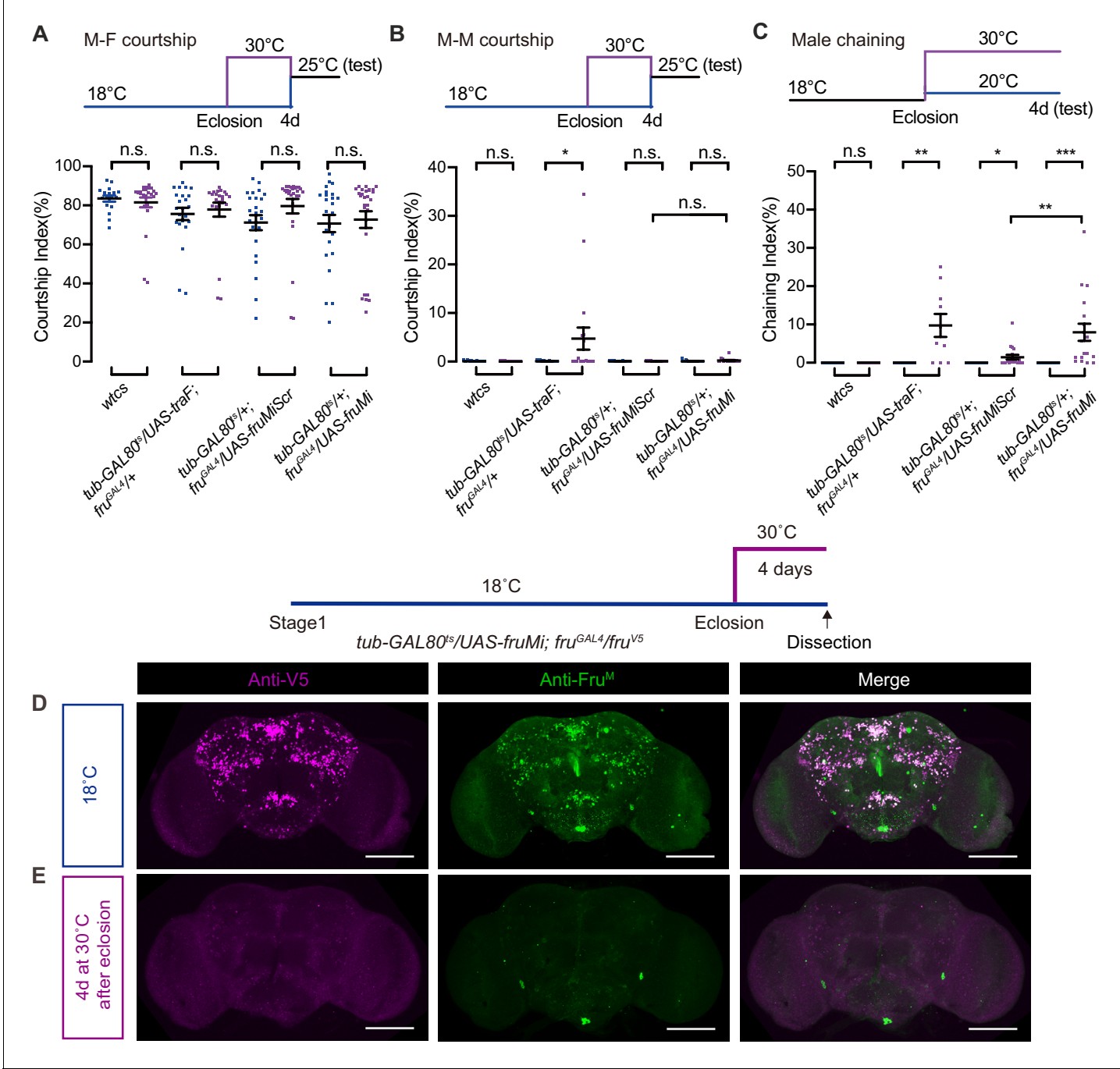

Figure 2. *fru^M* functions during adulthood to inhibit male–male courtship behaviors. (A–C) Courtship behaviors performed by males that express *traF* or *fruMi* specifically during adulthood for 4 days. For male–female courtship (A), n = 17, 26, 23, 23, 24, 27, 24, and 28, respectively (from left to right), n. s., not significant, unpaired t-test. For single-pair male–male courtship (B), n = 18 for each. n.s., not significant, *p<0.05, unpaired t-test. For male chaining among eight males as a group (C), n = 8, 8, 8, 10, 8, 18, 8, and 18, respectively (from left to right). n.s., not significant, *p<0.05, **p<0.01, ***p<0.001, Mann–Whitney U test. Error bars indicate SEM. Genotypes as indicated. (D and E) Anti-V5 and anti-Fru^M signals are dramatically decreased after heat shock during adulthood for 4 days in *tub-GAL80^ts/UAS-fruMi; fru^GAL4/fru^V5* males. Scale bars, 100 µm. Representative of five samples each. The online version of this article includes the following source data for figure 2:

**Source data 1.** Source data for *Figure 2*.

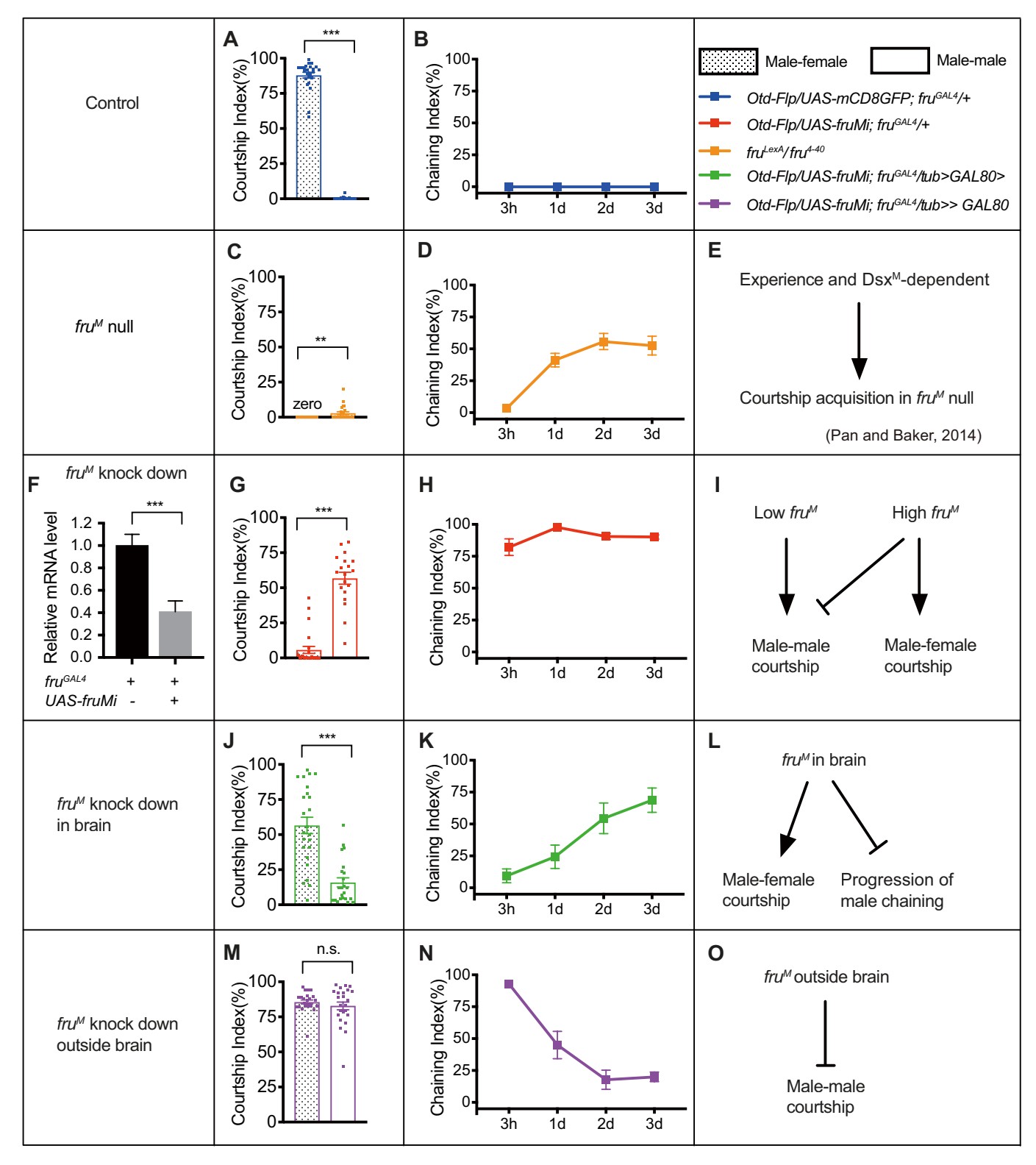

**Figure 3.** *fru^M* tunes functional flexibility of the *fru^M* circuitry. (A and B) Wild-type males courted intensively toward virgin females (A, left bar), but rarely courted males (A, right bar) or displayed chaining behavior in groups of eight males (B). n = 24, 24, 8, respectively. ***p<0.001, unpaired t-test. (C) *Fru^LexA^/fru^4-40^* (*fru^M* null) males rarely courted either females or males. n = 24 for each, **p<0.01, Mann–Whitney U test. (D) *Fru^LexA^/fru^4-40^* males did not show chaining behavior after 3 hr group-housing, but developed intensive chaining behavior after 1-3 days. n = 8. (E) A summary of courtship

*Figure 3 continued on next page*

*Figure 3 continued*

acquisition independent of *fru*$^M$. (**F**) RNAi against *fru*$^M$ efficiently decreased but not fully eliminated *fru*$^M$ expression. n = 4. ***p<0.001, Mann–Whitney U test. (**G**) Knocking down *fru*$^M$ in all *fru*$^{GAL4}$ neurons generated males that have reversed sexual orientation such that they rarely courted females but intensively courted males. n = 24 and 19, respectively. ***p<0.001, unpaired t-test. (**H**) Males with *fru*$^M$ knocked down in all *fru*$^{GAL4}$ neurons showed intensive chaining behavior at all time points (from 3 hr to 3 days upon group-housing). n = 7. (**I**) Distinct roles of low *fru*$^M$ (RNAi) and high *fru*$^M$ (wild-type) in regulating male–male and male–female courtship. (**J**) Males with *fru*$^M$ knocked down in *fru*$^{GAL4}$ neurons in the brain had a lower level of courtship toward females, but their sexual orientation was not changed. n = 24 and 23, respectively. ***p<0.001, unpaired t-test. (**K**) Males with *fru*$^M$ knocked down in *fru*$^{GAL4}$ neurons in brain showed low male chaining behavior initially but increasing levels of chaining behavior over 1–3 days. n = 6. (**L**) A summary of the role of *fru*$^M$ in brain in promoting male–female courtship and suppressing the experience-dependent acquisition or progression of male chaining behavior. (**M**) Males with *fru*$^M$ knocked down in *fru*$^{GAL4}$ neurons outside brain generated bisexual males that have intensive male–female and male–male courtship. n = 24 for each. n.s., not significant, unpaired t-test. (**N**) Males with *fru*$^M$ knocked down in *fru*$^{GAL4}$ neurons outside brain showed high male chaining behavior initially, but decreased levels of chaining behavior over 1–3 days. n = 8. (**O**) A summary of the role of *fru*$^M$ outside brain in suppressing male–male courtship behavior. Error bars indicate SEM.

The online version of this article includes the following source data and figure supplement(s) for figure 3:

**Source data 1.** Source data for *Figure 3*.
**Figure supplement 1.** Comparison of male courtship with or without food.
**Figure supplement 1—source data 1.** Source data for *Figure 3—figure supplement 1*.
**Figure supplement 2.** Courtship behaviors in *fru*$^M$ null males.
**Figure supplement 2—source data 1.** Source data for *Figure 3—figure supplement 2*.
**Figure supplement 3.** Dividing *fru*$^M$ expression into two complementary parts.
**Figure supplement 4.** The role of *fru*$^M$ in P1 and *ppk23*-expressing neurons for male courtship.
**Figure supplement 4—source data 1.** Source data for *Figure 3—figure supplement 4*.

exists a *fru*$^M$-independent experience and *dsx*$^M$-dependent courtship pathway (*Pan and Baker, 2014*; *Figure 3E*). To compare behavioral differences by *fru*$^M$ null males and *fru*$^M$ RNAi knocked down males that have systemic low level of *fru*$^M$, we firstly quantified to how much extent the micro-RNA against *fru*$^M$ (*UAS-fruMi* at attp40) worked. We found that the *fru*$^M$ mRNA level was reduced to ~40% of that in control males (*Figure 3F*). Interestingly, while males with *fru*$^M$ knocked down in all *fru*$^M$ neurons rarely courted females (CI ~5%, *Figure 3G*), they displayed a high level of male–male courtship behavior (CI > 50%, *Figure 3G*) and constantly high level of male chaining (*Figure 3H*), dramatically different from *fru*$^M$ null males. These results reveal distinct roles of low *fru*$^M$ (RNAi) and high *fru*$^M$ (wild-type) in regulating male–male and male–female courtship (*Figure 3I*).

To further reveal the role of *fru*$^M$ expression patterns in determining male courtship modes, we tried to spatially knockdown *fru*$^M$ expression using a simple way: *fru*$^M$ in brain and *fru*$^M$ outside brain. We used *Otd-Flp* expressing FLP specifically in the central brain (*Asahina et al., 2014*) to divide *fru*$^{GAL4}$ expression (*Figure 3—figure supplement 3A*) into two parts: *fru*$^M$- and *Otd*-positive neurons (specifically in brain) in *Otd-Flp/UAS-mCD8GFP; fru*$^{GAL4}$*/tub>GAL80>* males (*Figure 3—figure supplement 3B*) and *fru*$^M$-positive but *Otd*-negative neurons (theoretically outside brain, but still with few in brain) in *Otd-Flp/UAS-mCD8GFP; fru*$^{GAL4}$*/tub>stop>GAL80* males (*Figure 3—figure supplement 3C*). We also checked GFP expression in peripheral nervous system in these males and found a few GFP-positive cells in antennae and forelegs in *Otd-Flp/UAS-mCD8GFP; fru*$^{GAL4}$*/+* males, but rare expression in *Otd-Flp/UAS-mCD8GFP; fru*$^{GAL4}$*/tub>stop>GAL80* or *Otd-Flp/UAS-mCD8GFP; fru*$^{GAL4}$*/tub>GAL80>* males (*Figure 3—figure supplement 3D,E*). Thus, we successfully divided *fru*$^{GAL4}$ expression into two categories: one with *GAL4* expressed in *fru*$^+$*Otd*$^+$ neurons in brain and the other with *GAL4* expressed in *fru*$^+$*Otd*$^-$ neurons outside brain. We then used the above intersectional strategy to specifically knockdown *fru*$^M$ expression in or outside brain. To validate such strategy, we used anti-V5 to visualize Fru$^M$ expression in these males (together with *fru*$^{V5}$) and found effective, if not perfect, knockdown of Fru$^M$ expression spatially (*Figure 3—figure supplement 3F–I*). We found that males with *fru*$^M$ knocked down specifically in brain had a reduced level of courtship toward females (CI = 56.61 ± 5.86%), but their sexual orientation was not changed as they courted males in a much lower level (CI = 15.94 ± 3.26%, *Figure 3J*). Furthermore, males with *fru*$^M$ knocked down in brain showed low male chaining behavior initially but increasing levels of chaining behavior over 1–3 days (ChI [3 hr] = 9.35 ± 5.40%, ChI[3d] = 68.82 ± 5.53%, *Figure 3K*). Knocking down *fru*$^M$ only in a subset of male-specific P1 neurons driven by *P1-splitGAL4* in the brain that are important for courtship initiation (*Clowney et al., 2015*; *Kallman et al., 2015*;

*Kimura et al., 2008*; *Pan et al., 2012*; *Wu et al., 2019*) failed to decrease male–female courtship or induce male chaining behavior (*Figure 3—figure supplement 4A,B*). These results indicate that $fru^M$ function in brain promotes male–female courtship and inhibits acquisition or progression of the experience-dependent chaining behavior (*Figure 3L*). In contrast, males with $fru^M$ knocked down outside brain showed equally intensive male–female and male–male courtship (CI [male–female] = 85.62 ± 1.42%, CI [male–male] = 82.89 ± 2.76%, *Figure 3M*), indicating an inhibitory role of $fru^M$ in these neurons for male–male courtship (*Figure 3O*). These males performed a high level of male chaining behavior initially (ChI [3 hr] = 92.90 ± 3.08%), but decreased levels of chaining behavior over 1–3 days (ChI [3d] = 20.01 ± 3.75%, *Figure 3N*), consistent with the above finding that $fru^M$ function in the brain which is intact in these males inhibits acquisition or progression of male chaining behavior (*Figure 3L*). Knocking down $fru^M$ in a subset of gustatory receptor neurons expressing *ppk23* that respond to female-specific pheromones (*Lu et al., 2012*; *Thistle et al., 2012*; *Toda et al., 2012*) mildly enhanced male–male courtship but did not induce male chaining behavior (*Figure 3—figure supplement 4C,D*), suggesting a moderate role of $fru^M$ in these neurons for inhibiting male–male courtship, although its roles in these neurons during development or adulthood were not yet discriminated.

Taken together, the above results demonstrate distinct roles of $fru^M$ expression during a critical developmental period for the manifestation of courtship behaviors and adulthood for inhibiting male–male courtship (*Figure 4A*), and further reveal that different $fru^M$ expression levels and patterns determine courtship modes, indicative of functional flexibility of the $fru^M$-expressing sex circuitry tuned by $fru^M$ function (*Figure 4B*).

## Discussion

Previous findings show that $fru^M$ expression commences at the wandering third-instar larval stage, peaks at the pupal stage, and thereafter declines but does not disappear after eclosion (*Lee et al., 2000*), which suggests that $fru^M$ may function mainly during development for adult courtship behavior despite of no direct evidence. Here we temporally knocked down $fru^M$ expression in different developmental stages for 2 days and found that males with $fru^M$ knocked down during pupation rarely courted, while males with $fru^M$ knocked down during adulthood courted normally toward females. This is the first direct evidence that $fru^M$ is required during development but not adulthood for female-directed courtship behavior. A caveat of these experiments is that while $fru^M$ expression is effectively knocked down upon 2 day induction of $fru^M$ microRNA, it is not restored acutely after transferring to permissive temperature, although it is restored in adulthood if induction of $fru^M$ microRNA was performed at earlier stages (stages 1–5). Such a caveat does not compromise the above conclusion as knocking down $fru^M$ during pupation (stage 5) almost eliminated male courtship while knocking down at later stages have minor or no effect on male courtship. Consistent with these behavioral findings, knocking down $fru^M$ during stages 5 and 6, but not later stages, results in developmental defect in the gustatory receptor neurons innervating VNC.

In addition to the role of $fru^M$ during development to specify female-directed courtship, we also found a role of $fru^M$ during adulthood in suppressing male–male courtship, as males with $fru^M$ knocked down or *tra* overexpressed during adulthood displayed enhanced male–male courtship or male chaining behaviors. Note that a previous study found that removal of *transformer 2* (*tra2*) specifically during adulthood using a temperature sensitive *tra2* allele induced 8 of 96 females to show male-type courtship behaviors (*Belote and Baker, 1987*), which suggests that expression of FRU$^M$ and DSX$^M$ (by removal of *tra2* function in females) during adulthood is sufficient to masculinize CNS to some extent and induce a small fraction of females to display male-like courtship behaviors. Recent studies also found that $fru^M$ expression in the *Or47b*-expressing olfactory receptor neurons as well as their neuronal sensitivity depend on social experiences during adulthood (*Hueston et al., 2016*; *Sethi et al., 2019*). Based on all these findings, we propose that $fru^M$ expression during pupation is crucial for neuronal development and reconstruction of adult sex circuitry that allows innate courtship toward females, and its expression during adulthood may be activity dependent in at least some neurons and modulates some aspects of courtship (e.g., inhibits male–male courtship). Thus, there are at least two separate mechanisms that $fru^M$ contributes to the sex circuitry, one during a critical developmental period to build the female-directed innate courtship into that circuitry, and the other during adulthood to modulate neuronal physiology in an experience-dependent manner.

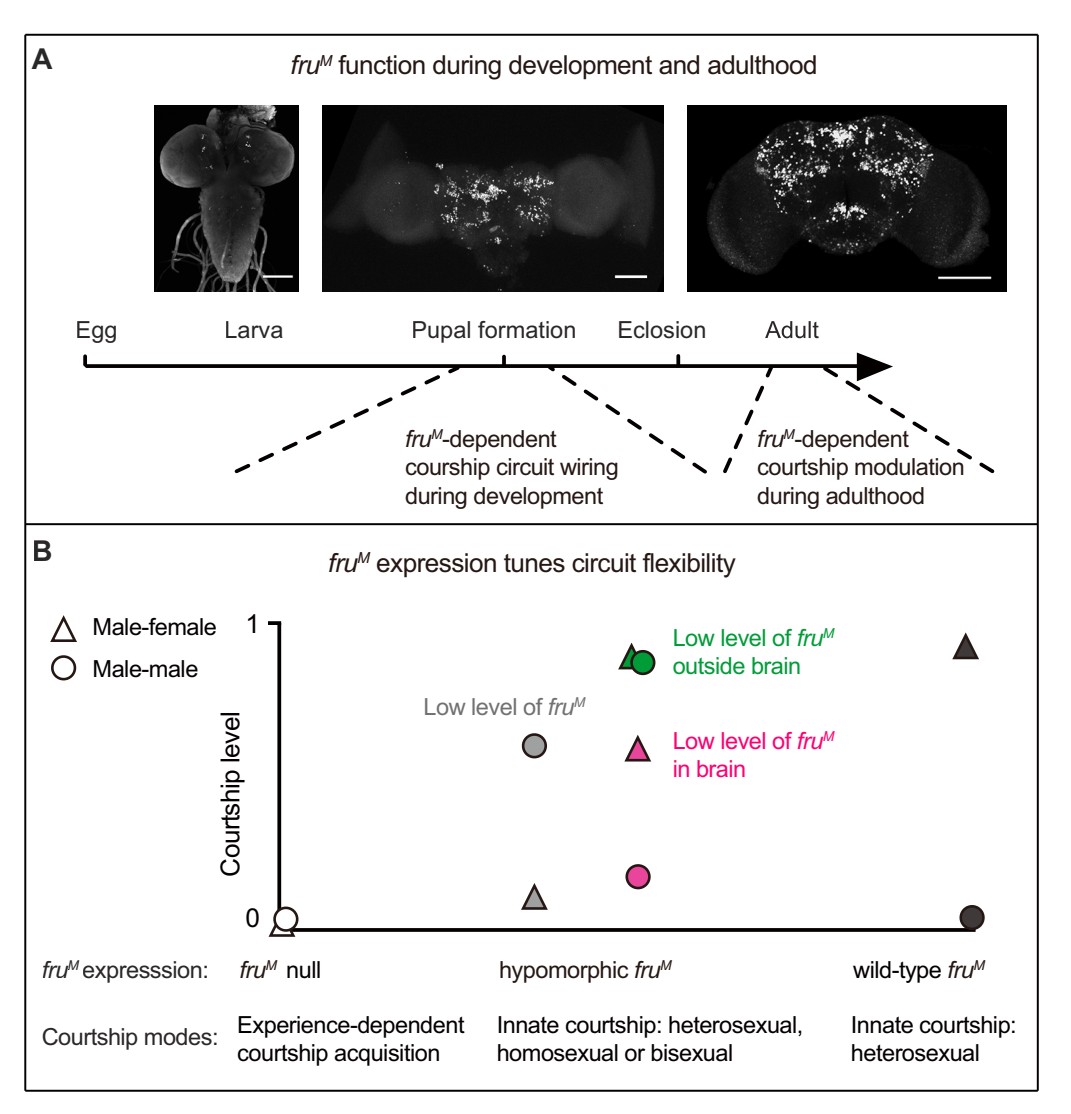

**Figure 4.** A summary of $fru^M$ function in male courtship. (**A**) $Fru^M$ is required during pupation for neuronal development and possibly circuit wiring that builds the potential for innately female-directed courtship, while its function during adulthood is involved in inhibiting male–male courtship. Anti-V5 signals indicate $Fru^M$ expression in larva, pupa and adult males (from left to right). Scale bars, 100 μm. (**B**) The sex circuitry without $fru^M$ or with different levels/patterns of $fru^M$ has different properties such that males would have experience-dependent courtship acquisition, or innate courtship but with different sexual orientation (heterosexual, homosexual, or bisexual). Such flexibility of the sex circuitry is tuned by different $fru^M$ expression. Triangles and circles represent corresponding $fru^M$ levels and courtship levels (triangles: male–female courtship; circles: male–male courtship). Gray indicates systemic low level of $fru^M$; green and magenta indicate spatially low level of $fru^M$.

Most importantly, we revealed striking flexibility of the fly sex circuitry by manipulating $fru^M$ expression. We listed four cases with $fru^M$ manipulation here for comparison: (1) males with a sex circuitry having wild-type $fru^M$ function have innate heterosexual courtship, as they court readily toward females, but do not court males no matter how long they meet; (2) males with a sex circuitry having no $fru^M$ function lose the innate courtship ability, but have the potential to acquire courtship toward males, females, and even other species in an experience-dependent manner; (3) males with a sex circuitry having limited $fru^M$ expression (e.g., 40%) have innate homosexual courtship, as they court readily toward other males, but rarely court females; (4) males with a sex circuitry having limited $fru^M$ expression outside brain (but intact $fru^M$ expression in brain) are innately bisexual, as they court

equally toward females or males. Although previous studies found that different $fru^M$ alleles (e.g., deletions, inversions, or insertions related to *fru*) showed very different courtship abnormalities (*Anand et al., 2001*; *Villella et al., 1997*), it was very hard to link $fru^M$ function to the flexibility of sex circuitry and often seen as allele-specific or background-dependent phenotypes. Our study using relatively simple genetic manipulations that generate dramatical different courtship behaviors promoted us to speculate a different view about the role of $fru^M$: instead of simply being a master gene that controls all aspects of male courtship, $fru^M$ is not absolutely necessary for courtship, but changes the wiring of the sex circuitry during development such that the sex circuitry may function in very different ways, ranging from innately heterosexual, homosexual, bisexual, to largely experience-dependent acquisition of the behavior. Such flexibility of the sex circuitry is tuned by different $fru^M$ expression, such that changes of $fru^M$ regulatory regions during evolution would easily select a suitable functional mode of the sex circuitry.

# Materials and methods

## Key resources table

| Reagent type (species) or resource | Designation | Source or reference | Identifiers | Additional information |
|---|---|---|---|---|
| Antibody | Mouse monoclonal anti-Bruchpilot antibody (nc82) | Developmental Studies Hybridoma Bank | Cat# nc82, RRID:AB_2314866 | IHC (1:50) |
| Antibody | Rabbit polyclonal anti-GFP | Thermo Fisher Scientific | Cat# A-11122, RRID:AB_221569 | IHC (1:1000) |
| Antibody | Donkey polyclonal anti-Rabbit, Alexa Fluor 488 | Thermo Fisher Scientific | Cat# A-21206, RRID:AB_2535792 | IHC (1:500) |
| Antibody | Donkey polyclonal anti-Mouse, Alexa Fluor 555 | Thermo Fisher Scientific | Cat# A-31570, RRID:AB_2536180 | IHC (1:500) |
| Antibody | Mouse monoclonal anti-V5-Tag:DyLight550 | Bio-Rad | Cat# MCA1360D550GA, RRID:AB_2687576 | IHC (1:500) |
| Antibody | Rabbit polyclonal anti-Fru$^M$ | This study | N/A | IHC (1:200) |
| Plasmid | pCFD4 | Addgene | # 49411 | |
| Plasmid | pHD-DsRed | Addgene | # 51434 | |
| Plasmid | pET-28a | Sigma–Aldrich | # 69864 | |
| Chemical compound, drug | Normal Goat Serum (NGS) | Jackson ImmunoResearch Laboratories | Code# 005-000-121 RRID:AB_2336990 | |
| Chemical compound, drug | Paraformaldehyde (PFA) | Sigma–Aldrich | CAS# 30525-89-4 | 4% PFA in 1× PBS |
| Genetic reagent (*D. melanogaster*) | $fru^{V5}$ | This study | N/A | Described below |
| Genetic reagent (*D. melanogaster*) | *UAS-mCD8GFP; fru$^{GAL4}$* | *Stockinger et al., 2005* | N/A | |
| Genetic reagent (*D. melanogaster*) | *UAS-fruMi* | *Meissner et al., 2016* | N/A | |
| Genetic reagent (*D. melanogaster*) | *UAS-fruMiScr* | *Meissner et al., 2016* | N/A | |
| Genetic reagent (*D. melanogaster*) | $fru^{LexA}$ | *Mellert et al., 2010* | N/A | |

*Continued on next page*

*Continued*

| Reagent type (species) or resource | Designation | Source or reference | Identifiers | Additional information |
|---|---|---|---|---|
| Genetic reagent (*D. melanogaster*) | *fru$^{4-40}$* | *Pan and Baker, 2014* | N/A | |
| Genetic reagent (*D. melanogaster*) | *fru$^{Sat15}$* | *Pan and Baker, 2014* | N/A | |
| Genetic reagent (*D. melanogaster*) | *fru$^{AJ96u}$* | *Pan and Baker, 2014* | N/A | |
| Genetic reagent (*D. melanogaster*) | *ppk23-GAL4* | *Thistle et al., 2012* | N/A | |
| Genetic reagent (*D. melanogaster*) | *Otd-Flp* | *Asahina et al., 2014* | N/A | |
| Genetic reagent (*D. melanogaster*) | *tub-GAL80$^{ts}$* | Bloomington Drosophila Stock Center | BDSC_7019 | |
| Genetic reagent (*D. melanogaster*) | *tub>GAL80>* | Bloomington Drosophila Stock Center | BDSC_38881 | |
| Genetic reagent (*D. melanogaster*) | *tub>stop>GAL80* | Bloomington Drosophila Stock Center | BDSC_39213 | |
| Genetic reagent (*D. melanogaster*) | *UAS-traF* | Bloomington Drosophila Stock Center | BDSC_4590 | |
| Genetic reagent (*D. melanogaster*) | *R15A01-AD* | Bloomington Drosophila Stock Center | BDSC_68837 | |
| Genetic reagent (*D. melanogaster*) | R71G01-DBD | Bloomington Drosophila Stock Center | BDSC_69507 | |
| Software, algorithm | ImageJ | National Institutes of Health | https://imagej.nih.gov/ij/ | |
| Software, algorithm | Prism 8 | GraphPad | https://www.graphpad.com/ | |

## Fly stocks

Flies were maintained at 22 or 25°C in a 12 hr:12 hr light:dark cycle. Canton-S flies were used as the wild-type strain. Other stocks used in this study include the following: *fru$^{GAL4}$* (*Stockinger et al., 2005*), *fru$^{V5}$* (this study), *UAS-fruMi* (attp40), *UAS-fruMi* (attp2), and *UAS-fruMiScr* (attp2) (*Meissner et al., 2016*), *fru$^{LexA}$*, *fru$^{4-40}$*, *fru$^{AJ96u3}$*, and *fru$^{Sat15}$* (*Pan and Baker, 2014*), *ppk23-GAL4* (*Thistle et al., 2012*), *P1-splitGAL4* (*R15A01-AD; R71G01-DBD*) (*Zhang et al., 2018*), and *Otd-Flp* (*Asahina et al., 2014*). *UAS-traF* (BL#4590), *tub-GAL80$^{ts}$* (BL#7019), *tub>GAL80>* (BL#38881), and *tub>stop>GAL80* (BL#39213) were from Bloomington Drosophila Stock Center.

## Generation of *fru$^{V5}$*

*fru$^{V5}$* was generated by fusing V5 tag in frame with the start codon of *fruP1*. To generate the *fru$^{V5}$* knock-in line, two gRNAs (gRNA1: 5′-GCCATTAGTGTCGCGGTGCG-3′; gRNA2: 5′-GCGGCCGCGCGAGTCGCCGC-3′) against *fru* were inserted into pCFD4 vector (Addgene #49411) to induce DNA break near the start codon of *fruP1*. Then, ~2.1 kb 5′ homologous arm was incorporated into the 5′ MCS of pHD-DsRed (Addgene #51434) through Gibson assembly (digested with NheI and NdeI). To insert V5 tag after the start codon of *fruP1*, ~2.4 kb 3′ homologous arm was divided into two fragments and amplified separately. These two fragments including the V5 sequence were then subcloned into the 3′ MCS of pHD-DsRed (containing the above 5′ homologous arm) through Gibson assembly (digested with BglII and XhoI). The modified pCFD4 and pHD-DsRed

plasmids were injected into *vas-cas9* embryos. Successful knock in was selected by 3xP3-DsRed (DsRed-positive eyes) and confirmed by PCR followed by sequencing. The verified knock-in line was balanced and crossed to *hs-Cre* flies to remove the 3xP3-DsRed marker.

## Generation of anti-Fru$^M$ antibody

The rabbit polyclonal antibody against Fru$^M$ was generated by ABclonal (Wuhan, China). In brief, the fragment of *fru* gene encodes the N-terminal 101 amino acids, starting with MMATSQDYFG and ending in SPRYNTDQGA, was cloned into expression vector pET-28a (Sigma–Aldrich, #69864). The 101 amino acids are only present in male-specific Fru proteins (Fru$^M$) from *fruP1*. A SUMO-tagged Fru$^M$ fusion antigen was synthesized from bacteria, purified, and used to immunize a rabbit. The anti-Fru$^M$ antibody was affinity purified.

## Courtship and chaining assays

For the single-pair courtship assay, the tester males and target flies (4–8 days old) were gently aspirated into round two-layer chambers (diameter: 1 cm; height: 3 mm per layer) and were separated by a plastic transparent barrier that was removed ~30 min later to allow courtship test. Courtship index (CI), which is the percentage of observation time a fly performs any courtship step, was used to measure courtship to female targets or between two males. Paired male–male courtship used two males of the same genotype but focused on the male fly that first initiated courtship (courtship of the initiator to the other). All tester flies were single housed if not otherwise mentioned. Each test was performed for 10 min.

For male chaining assay, tester males (4–8 days old) were loaded into large round chambers (diameter: 4 cm; height: 3 mm) by cold anesthesia. Tests were performed daily for four consecutive days (3 hr after grouping as day 0, then days 1–3). For chaining behavior in *Figure 2C*, flies were only tested after grouping together for 3 days. Chaining index (ChI), which is the percentage of observation time at least three flies engaged in courtship together, was used to measure courtship in groups of eight males.

To generate males with *fru$^M$* knocked down only for 2 days during development or adulthood, we raised *tub-GAL80$^{ts}$/+; fru$^{GAL4}$/UAS-fruMi* flies at 18°C and transferred these flies to fresh food vials every 2 days. In this way, we generated *tub-GAL80$^{ts}$/+; fru$^{GAL4}$/UAS-fruMi* flies at nine different stages from embryos (stage 1) to newly eclosed adults (stage 9), with wandering larvae at stage 5 and early pupas at stage 6. We then transferred all these flies to a 30°C incubator allowing *fru$^M$* knockdown for 2 days, then placed all flies back to 18°C until courtship test at adult.

## Quantitative real-time PCR

Total RNA was extracted from ~15 male flies with TRIzol (15596026, Invitrogen), according to the manufacturer's instructions. The cDNA was synthesized using Prime Script reagent kit (18091050, Invitrogen). Quantitative PCR was performed on LightCycler 96 Real-Time PCR System (Roche) using AceQ qPCR SYBR Green Master Mix (Q121-02, Vazyme). *Actin* was used as control for normalization. The primers used were as follows: *Actin* (forward: 5′- CAGGCGGTGCTTTCTCTCTA-3′; reverse: 5′-AGCTGTAACCGCGCTCAGTA-3′), *fru* P1 promotor (forward: 5′-GTGTGCGTACGTTTGAGTGT-3′; reverse: 5′-TAATCCTGTGACGTCGCCAT-3′), and *fru* P4 promotor (forward: 5′-TGTATAGCGG-CAACTGAACC-3′; reverse: 5′-CCGGTCAAATTTGTGGGATG-3′).

## Immunohistochemistry

We dissected brains and ventral nerve cords of males in defined developmental stages (e.g., *Figure 1E–H*) or 5–7 days old males in Schneider's insect medium (Thermo Fisher Scientific, Waltham, MA) and fixed in 4% paraformaldehyde in phosphate-buffered saline (PBS) for 30 min at room temperature. After washing four times in 0.5% Triton X-100% and 0.5% bovine serum albumin [BSA] in PBS (PAT), tissues were blocked in 3% normal goat serum (NGS) for 60 min, then incubated in primary antibodies diluted in 3% NGS for ~24 hr at 4°C, washed (4× 15 min) in PAT at room temperature, and incubated in secondary antibodies diluted in 3% NGS for ~24 hr at 4°C. Tissues were then washed (4× 15 min) in PAT and mounted in Vectorshield (Vector Laboratories, Burlingame, CA) for imaging. Primary antibodies used were rabbit anti-Fru$^M$ (1:200; this study), mouse anti-V5-Tag: DyLight550 (1:500; MCA1360D550GA, Bio-Rad, Hercules, CA), rabbit anti-GFP (1:1000; A11122,

Invitrogen, Waltham, MA), and mouse anti-Bruchpilot (1:50; nc82, Developmental Studies Hybridoma Bank, Iowa City, IA). Secondary antibodies used were donkey anti-mouse IgG conjugated to Alexa 555 (1:500, A31570, Invitrogen) and donkey anti-rabbit IgG conjugated to Alexa 488 (1:500, A21206, Invitrogen). Samples were imaged at 10× or 20× magnification on a Zeiss 700 confocal microscope and processed with ImageJ.

### Statistics

Experimental flies and genetic controls were tested at the same condition, and data are collected from at least two independent experiments. Statistical analysis is performed using GraphPad Prism and indicated inside each figure legend. Data presented in this study were first verified for normal distribution by D'Agostino–Pearson normality test. If normally distributed, Student's t test is used for pairwise comparisons, and one-way ANOVA is used for comparisons among multiple groups, followed by Tukey's multiple comparisons. If not normally distributed, Mann–Whitney U test is used for pairwise comparisons, and Kruskal–Wallis test is used for comparisons among multiple groups, followed by Dunn's multiple comparisons.

## Acknowledgements

We thank the Bloomington Drosophila Stock Center for fly stocks. This work was supported by grants from National Key R and D Program of China (2019YFA0802400), the National Natural Science Foundation of China (31970943, 31622028, and 31700905), and the Jiangsu Innovation and Entrepreneurship Team Program.

## Additional information

### Funding

| Funder | Grant reference number | Author |
| --- | --- | --- |
| National Natural Science Foundation of China | 31970943 | Yufeng Pan |
| National Natural Science Foundation of China | 31700905 | Qionglin Peng |
| National Natural Science Foundation of China | 31622028 | Yufeng Pan |

The funders had no role in study design, data collection and interpretation, or the decision to submit the work for publication.

### Author contributions

Jie Chen, Conceptualization, Data curation, Formal analysis, Investigation, Methodology, Writing - original draft; Sihui Jin, Data curation, Investigation; Dandan Chen, Investigation, Methodology; Jie Cao, Data curation, Methodology; Xiaoxiao Ji, Investigation, Writing - review and editing; Qionglin Peng, Investigation, Methodology, Writing - review and editing; Yufeng Pan, Conceptualization, Supervision, Funding acquisition, Writing - original draft, Writing - review and editing

### Author ORCIDs

Yufeng Pan  https://orcid.org/0000-0002-1535-9716

### Decision letter and Author response

Decision letter https://doi.org/10.7554/eLife.59224.sa1
Author response https://doi.org/10.7554/eLife.59224.sa2

## Additional files

### Supplementary files
• Transparent reporting form

### Data availability
All data generated or analysed during this study are included in the manuscript and supporting files. Source data files have been provided for Figures 1, 2, 3, Figure 3-figure supplement 1, 2 and 4.

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
