## [Decision Letter]

**Acceptance summary:**

This study provides insight into how the sexually dimorphic transcription factor (Fruitless M) acts at different developmental stages to establish sex-specific courtship behavior. The novelty of the paper is that it identifies two distinct phases of the Fru^M^ action, one during the pupal stage and the other after adult emergence; the former is pivotal for normal development of female-directed courtship whereas the latter for suppressing male-to-male courtship.

**Decision letter after peer review:**

Thank you for submitting your article "fruitless tunes functional flexibility of courtship circuitry during development" for consideration by *eLife*. Your article has been reviewed by three peer reviewers, and the evaluation has been overseen by Kristin Scott as the Reviewing Editor and K VijayRaghavan as the Senior Editor. The reviewers have opted to remain anonymous.

The reviewers have discussed the reviews with one another and the Reviewing Editor has drafted this decision to help you prepare a revised submission.

Summary:

This short paper by Chen et al. is a crisp and clear study that provides evidence for spatial and temporal differences in the developmental and behavioral function of male isoform of fruitless (Fru^M^) in courtship. The authors found two distinct phases of the Fru^M^ action, one during the pupal stage and the other after adult emergence; the former is pivotal for normal development of female-directed courtship whereas the latter for suppressing male-to-male courtship. The authors refer to these two phases as innate and experience-dependent components of male courtship, respectively. Knockdown and mutations of fru^M^ were found to affect differently the innate and experience-dependent components, reflecting the different levels of residual fru^M^ functions. Although the role of Fru^M^ function is heavily studied in the context of male behaviors, the novelty of the paper is that it unravels spatial, temporal and quantitative segregation of Fru^M^ function in development of circuits driving courtship behaviors and its function in regulating innate reproductive behaviors. However, validation of the efficacy of fru^M^ manipulations is required in order to interpret the studies.

Essential revisions:

1) All of the conclusions critically depend on the assumption that RNAi-based knockdown of fru^M^ is effective. It is important to verify that RNAi does knock down fru^M^ expression in a developmental stage- or tissue-specific manner. Moreover, It is important to show that fruMi selectively inactivates fru^M^ without affecting non-sex specific transcripts by performing qPCR with primers specific for fru^M^ and Fru common forms. Immunohistochemistry (IHC), RNA in situ hybridization, or showing how knocking down fru at stage 5 changes the circuitry (i.e. numbers of fru positive cells or their wiring patterns) are approaches that may supplement as validation. Please provide details of qPCR or IHC.

2) In the experiment of stage-specific knockdown, no behavioral effect was observed when *GAL80* was inactivated before stage-4, letting the authors conclude that "the critical period" commences after stage-5. However, it is not certain whether fru-GAL4 was transcriptionally active at stage-4 or earlier stages. Images of tissues stained for fru-GAL4 activity along the developmental stage should be added in a supplementary figure.

3) As a fru^M^ null genotype, only fru[LexA]/fru[4-40] was used. More allelic combinations, those without fru[LexA] in particular, are to be tested, because fru[LexA] behaves differently from other alleles when male chaining is measured.

[Editors' note: further revisions were suggested prior to acceptance, as described below.]

Thank you for re-submitting your work entitled "fruitless tunes functional flexibility of courtship circuitry during development" for consideration by *eLife*. Your article has been reviewed by one peer reviewer who reviewed the original manuscript, and the evaluation has been overseen by a Reviewing Editor and a Senior Editor. The reviewer has opted to remain anonymous.

We regret to inform you that your work will not be considered further for publication in *eLife*. Specifically, the lack of validation of the efficacy of RNAi at different developmental stages was a major weakness noted in the original reviews and has not been addressed in the revision.

Major Comments:

The authors did not incorporate our request to verify the effect of RNAi "in a developmental stage- or tissue-specific manner" (that is, under conditions they employed in Figure 1). To reply to their arguments:

1) The new Figure 1—figure supplement 1B uses the constitutive expression of miRNA against fru^M^, just like data in Figure 2F. These are insufficient to address our concern that the authors' conclusion critically depends on unproven efficacy of stage-specific RNAi using *GAL80^ts^*. My request was simple: perform qRNA or immunohistochemistry against fru^M^ using exactly the same animals authors used in Figure 1.

2) One image showing neural development defect (Figure 1—figure supplement 1M) is inappropriate to prove the efficacy of RNAi treatment, for three reasons. First is an obvious lack of biological replicates. Second is that we still do not know whether the effect is generalizable across the entire nervous system. Lastly, if they wish to insist that the developmental defects caused by RNAi was the reason why their courtship performance plummeted at stage 5, they must perform similar comparison across stages. That is what we initially meant by verifications in "developmental stage-specific manner". What happens when knockdown is induced at stage 4 or stage 6? Does the anatomical phenotype correlate with behavior? This information must be critical for supporting the model authors shown in Figure 3.

3) GFP expression patterns (Figure 1—figure supplement 1D-K) does not replace the need to directly address the effects of RNAi. They should have performed direct evaluation of fru expression when RNAi was induced at stage 4, since their conclusion depends on the assumption that RNAi knocked fru^M^ down then.

Therefore, I am afraid authors have not presented sufficient data to support their conclusion regarding the developmental stage-specific effects of fru. I am particularly disappointed because 1) the experiment was straightforward and within the expertise of the authors, and 2) stage-specific function of fru^M^ takes up such a significant portion of Discussion.

I acknowledge that authors improved their manuscript by adding discussion and method details. Knowing the difficult situation that all scientists around the world are currently experiencing, I am very reluctant to be unnecessarily critical. However, the caveats on the verification of stage-specific RNAi are left largely unaddressed and are a major confound with the study.

---

## [Author Response]

Essential revisions:1) All of the conclusions critically depend on the assumption that RNAi-based knockdown of fru^M^ is effective. It is important to verify that RNAi does knock down fru^M^ expression in a developmental stage- or tissue-specific manner. Moreover, It is important to show that fruMi selectively inactivates fru^M^ without affecting non-sex specific transcripts by performing qPCR with primers specific for fru^M^ and Fru common forms. Immunohistochemistry (IHC), RNA in situ hybridization, or showing how knocking down fru at stage 5 changes the circuitry (i.e. numbers of fru positive cells or their wiring patterns) are approaches that may supplement as validation. Please provide details of qPCR or IHC.

We used *UAS-fruMi* (attp40) and *UAS-fruMi* (attp2) targeting the male-specific *fru^M^* mRNA, and a scrambled version *UAS-fruMiScr* (attp2) as a control. These stocks were generated previously in the Baker lab in Janelia and proved to be effective (Meissner et al., 2016; Chen et al., 2017). We now provided additional experiments validating the efficiency of these stocks. First, the *fru^M^* mRNA level was significantly lower using either *UAS-fruMi* (attp40) (Figure 2F) or *UAS-fruMi* (attp2) (Figure 1—figure supplement 1B) driven by *fru^GAL4^*. Furthermore, mRNA level of FruCom from the P4 promotor is not affected using *UAS-fruMi* driven by *fru^GAL4^* (Figure 1—figure supplement 1C). Knocking down *fru^M^* in all *fru^GAL4^* neurons may affect neuronal development of many types of *fru^M^* neurons and we focused on the gustatory receptor neurons innervating the VNC for their easily identified phenotype: loss of midline crossing (Figure 1—figure supplement 1M, white arrow). We found that knocking down *fru^M^* for 2 days at stage 5 is sufficient to result in loss of midline crossing of the gustatory receptor neurons in adult (Figure 1—figure supplement 1N, white arrow), further supporting the efficiency of *fru^M^* knock-down by *UAS-fruMi*.

2) In the experiment of stage-specific knockdown, no behavioral effect was observed when GAL80 was inactivated before stage-4, letting the authors conclude that "the critical period" commences after stage-5. However, it is not certain whether fru-GAL4 was transcriptionally active at stage-4 or earlier stages. Images of tissues stained for fru-GAL4 activity along the developmental stage should be added in a supplementary figure.

We provided *fru^GAL4^* expression images in all developmental stages from embryo to adult and found specific expression at all stages (Figure 1—figure supplement 1D-K).

3) As a fru^M^ null genotype, only fru[LexA]/fru[4-40] was used. More allelic combinations, those without fru[LexA] in particular, are to be tested, because fru[LexA] behaves differently from other alleles when male chaining is measured.

We did not provided additional *fru^M^* null mutant behaviors for the initial submission as there are already published data (Pan et al., 2014). We now re-tested two other *fru^M^* null combinational mutants (*fru^4-40^/fru^AJ96u3^* and *fru^4-40^/fru^Sat15^*) for their male-male and male-female courtship, as well as male chaining behavior (Figure 2—figure supplement 3A-F), which further supports our findings that *fru^M^* null males did not court either male or female flies initially, and developed intensive male chaining behavior after grouping for 1-3 days.

[Editors' note: further revisions were suggested prior to acceptance, as described below.]

Major Comments:The authors did not incorporate our request to verify the effect of RNAi "in a developmental stage- or tissue-specific manner" (that is, under conditions they employed in Figure 1). To reply to their arguments:1) The new Figure 1—figure supplement 1B uses the constitutive expression of miRNA against fru^M^, just like data in Figure 2F. These are insufficient to address our concern that the authors' conclusion critically depends on unproven efficacy of stage-specific RNAi using GAL80^ts^. My request was simple: perform qRNA or immunohistochemistry against fru^M^ using exactly the same animals authors used in Figure 1.

We only recently generated a *fru^V5^* knock-in line and anti-Fru^M^ antibody successfully, and these tools could allow us to visualize Fru^M^ expression to answer the above question. These tools were validated by immunostaining against V5 and Fru^M^ in wild-type males and females, as well as in *fru^M^* mutants (see Figure 1—figure supplement 2).

We then used the *fru^V5^* line and Fru^M^ antibody to check if 2-day heat shock at 30°C during development is sufficient to knock down *fru^M^* expression. We dissected brains immediately after 2-day heat shock at stage 5 or 7 and found anti-V5 and anti-Fru^M^ signals were dramatically decreased, such that only a small fraction of neurons could be weakly labelled. In comparison, control males with the same age but raised at 18˚C have regular anti-V5 and anti-Fru^M^ signals (see Figure 1). We actually performed such experiment from stage 4 to 9, all of which showed similarly strong knock-down of *fru^M^* expression, and we put the result of stage 5 and 7 into Figure 1 to show the efficiency of *fru^M^* microRNAi. We also checked anti-V5 and anti-Fru^M^ signals in males that have been heat shocked for 4 days during adulthood and found similar knock-down effect (now part of Figure 2). These results clearly demonstrate an effective knock-down of *fru^M^* expression upon 2-day heat shock at 30˚C.

As induction of *fru^M^* microRNA for 2 days effectively knocked down *fru^M^* expression during that period, but *fru^M^* microRNA may not be degraded immediately and has longer effect, we further tested to how much extent such knock-down effect may last. Thus, we dissected brains of adult males that have been heat shocked for 2 days at different developmental stages (from stage 1 to 9), and found that males that have been heat shocked at earlier stages (such as stage 1-5) still have strong Fru^M^ expression (check the anti-V5 signals, the anti-Fru^M^ signals are relatively weaker), while those heat shocked at later stages (stages 6-9) still have reduced Fru^M^ expression in adult males (see Figure 1—figure supplement 3).

Thus, these results showed that induction of *fru^M^* microRNA for 2 days effectively knocked down *fru^M^* expression, and Fru^M^ expression did not fully restore to regular levels if heat shocked at later stages. However, such a caveat (acknowledged in the discussion part) does not compromise our conclusion at all, as even if induction of *fru^M^* microRNA at stage 5 may have minor effect on later stages, its role on behavior is still due to stage 5, as induction of *fru^M^* microRNA at stages 6-9 did not affect male courtship.

As we now have anti-V5 to visualize Fru^M^, we also checked efficiency of spatial *fru^M^* knock-down in flies using *Otd-flp*. We found that consistent with *fru^GAL4^* expression (labeled by GFP), Fru^M^ was efficiently knocked down mainly in brain or VNC (see Figure 3—figure supplement 3).

For stage-dependent developmental deficit of gustatory receptor neurons, see below.

2) One image showing neural development defect (Figure 1—figure supplement 1M) is inappropriate to prove the efficacy of RNAi treatment, for three reasons. First is an obvious lack of biological replicates. Second is that we still do not know whether the effect is generalizable across the entire nervous system. Lastly, if they wish to insist that the developmental defects caused by RNAi was the reason why their courtship performance plummeted at stage 5, they must perform similar comparison across stages. That is what we initially meant by verifications in "developmental stage-specific manner". What happens when knockdown is induced at stage 4 or stage 6? Does the anatomical phenotype correlate with behavior? This information must be critical for supporting the model authors shown in Figure 3.

In order to address this concern, we tried to analyze the midline crossing defects of the gustatory receptor neurons (GRNs) in males that have been heat shocked for 2 days in each developmental stage, although we still could not analyze other neuronal types due to technical limits (hard to identify developmental defect of other neuronal types using the genetic setup in this study) as well as the limited scope of this study.

We first checked basal expression of GFP in *tub-GAL80^ts^/UAS-mCD8GFP; fruGAL4/UAS-fruMi* males raised constantly at 18°C, and observed a small fraction of neurons expressing GFP (see Figure 1—figure supplement 4), which indicates a strong but not 100% suppression of GAL4 activity by GAL80^ts^ at 18˚C. Note that the GRNs we tried to analyze were not GFP-positive, indicating a successful suppression of GAL4 activity in these neurons at 18˚C.

We then dissected VNC of *tub-GAL80ts/UAS-mCD8GFP; fruGAL4/UAS-fruMi* males that have been heat shocked for 2 days in each developmental stage, with 5 samples for each stage. We found that these GRNs were only labeled in males that have been heat shocked after stage 4 (stages 5-9, yellow arrows), probably because these GRNs were developed during pupation, which is consistent with a previous study (Mellert et al., 2012). Thus, we can only compare the effect of *fru^M^* knock down after stage 4 on the development of these GRNs.

Interestingly, we found that all males heat shocked at stage 5 showed defect of midline crossing in these GRNs, 60% of males heat shocked at stage 6 showed defect of midline crossing, while all males heat shocked after stage 6 showed regular midline crossing (white arrow). Males heat shocked for 4 days during adulthood also have regular midline crossing. These results clearly showed a critical developmental period (stages 5-6) where Fru^M^ function to ensure regular development of GRNs.

3) GFP expression patterns (Figure 1—figure supplement 1D-K) does not replace the need to directly address the effects of RNAi. They should have performed direct evaluation of fru expression when RNAi was induced at stage 4, since their conclusion depends on the assumption that RNAi knocked fruM down then.

We now performed RNAi in every stage and used anti-V5, anti-Fru^M^ and anti-GFP to address these concerns, as above described.

Therefore, I am afraid authors have not presented sufficient data to support their conclusion regarding the developmental stage-specific effects of fru. I am particularly disappointed because 1) the experiment was straightforward and within the expertise of the authors, and 2) stage-specific function of fru^M^ takes up such a significant portion of Discussion.I acknowledge that authors improved their manuscript by adding discussion and method details. Knowing the difficult situation that all scientists around the world are currently experiencing, I am very reluctant to be unnecessarily critical. However, the caveats on the verification of stage-specific RNAi are left largely unaddressed and are a major confound with the study.

We thank the reviewer for these comments, and we now performed extensive analysis of Fru^M^ expression using anti-V5 and anti-Fru^M^ antibodies, as well as developmental defect of GRNs, in males that have induced expression of *fru^M^* microRNAi in almost every developmental stage. We believe these results have addressed reviewer’s concerns and significantly enhanced this manuscript.